# Beachrock Morphology along the Mediterranean Coast of Israel: Typological Classification of Erosion Features

**Amir Bar [1], Revital Bookman [1], Ehud Galili [2,3] and Dov Zviely [4,*]**

[1]  Department of Marine Geosciences, The Leon H. Charney School for Marine Sciences, University of Haifa, Haifa 3498838, Israel
[2]  The Zinman Institute of Archaeology, University of Haifa, Haifa 3498838, Israel
[3]  The Recanati Institute for Maritime Studies, University of Haifa, Haifa 3498838, Israel
[4]  Faculty of Marine Sciences, Ruppin Academic Center, Emek-Hefer 40250, Israel
*   Correspondence: dovz@ruppin.ac.il; Tel.: +972-525-805-758

**Abstract:** Beachrock is composed of intertidal-associated sediments, rapidly cemented by calcium carbonate, and has important implications for understanding coastal morphological processes. This study focuses on the morphodynamic erosion patterns of Late Holocene beachrock outcrops along the Mediterranean coast of Israel that have formed since the sea reached its present level about 4000 years ago. Exposed beachrock is subjected to erosion, affecting its seaward and landward facing fronts and upper surface, and creating distinct morphological features due to wave pounding and coastal currents which remove unconsolidated sediment supporting layers. The current state of beachrock morphology is presented, based on field measurements and field relation interpretations of selected sites, backed by petrographic and sedimentological data. It shows and studies selected beachrock exposures along Israel's coast, and characterizes their morphological features in various field-relation configurations. A classification is developed of the main erosion patterns of beachrock embedded on loose, hard, partly hard and partly loose substrate.

**Keywords:** beachrock weathering; coastal morphology; Israeli coast; sandy beach; rocky platforms; coastal erosion; Holocene sea level

## 1. Introduction

Beachrock (BR) is a rocky formation characterized by the significant presence of marine-associated clasts, such as coarse sand and broken shells, rapidly cemented by calcium carbonate within the intertidal zone. Beachrock is relatively common, and most reported BR occurrences are on tropical and subtropical micro-tidal coasts [1–4]. A BR outcrop represents fossil bulk sediment (Figure 1A), consolidated while buried beneath the former beach face (the terminology used to describe the beach profile in Figure 1 is after [1]). Exposure of the buried BR in the swash zone is likely to occur at a later stage (Figure 1B,C), following the erosion of overlying unconsolidated sediments [5–9].

Beachrock outcrops usually indicate beach face erosion, however, once exposed they may protect the beach face from sediment loss [10–12]. The sedimentary particles and cements in BR reflect the nature of the unconsolidated sedimentary material at the coast, and are influenced by the local environmental conditions during diageneses [8,13,14].

Beachrock studies contribute greatly to the understanding of coastal environment evolution (e.g., [13–34]) and can be utilized as an indicator for reconstructing paleo-shorelines, relative sea-level change, and tectonic movements. In some cases, considerable sections of BR are not in-situ: at the sea bottom, 2–4 m below present sea level (bsl) and at the backshore, about 2–3 m above present sea level (asl). Incorrect interpretation of the BR morphological evolution can lead to a misunderstanding of sea-level change, vertical movements of the Earth's crust and shoreline migrations.

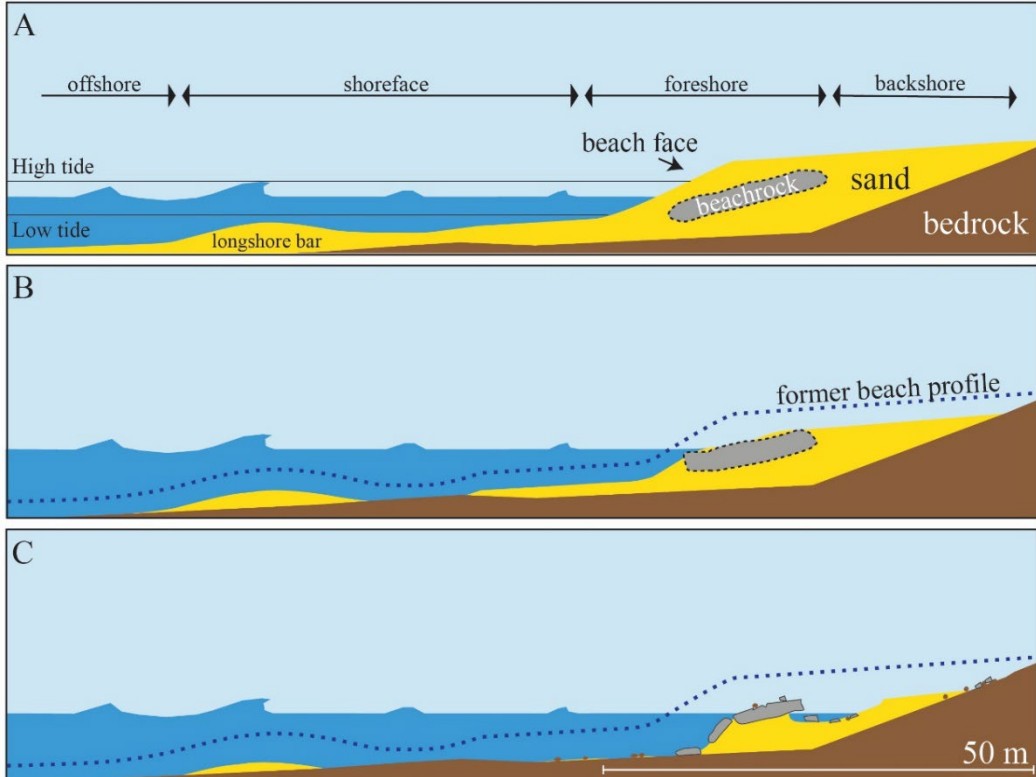

**Figure 1.** Schematic lateral cross section of a beach depicting stages in BR outcrop evolution inside an unconsolidated sediment lens: (**A**) initial stage: formation of solid sub-surface, BR accumulation within the shore face; (**B**) intermediate stage: erosion of sandy beach face and exposure of BR (the initial beach section is shown dotted); (**C**) progressive stage: extensive exposure of BR outcrop and development of erosion features.

In this article we present study cases of BR outcrop morphology along the Mediterranean coast of Israel, and use morphological observations to suggest a typological division of BR erosion features. Determination of the environmental mechanisms involved in the genesis stage of BR outcrops and for tracking changes in relative sea level (RSL) and sediment transportation, requires sedimentological analysis, and is beyond the scope of this paper. These observations (although not comprising all BR outcrops along the Israeli coast) are essential for the establishment of the geographical layout of BR along the Central and Carmel coasts (Figure 2), understanding the evolution of sandy beaches, and examining the use of BR as a proxy for natural processes and events.

### 1.1. The Study Area

The Mediterranean coastline of Israel extends 195 km from Zikim in the south, to Rosh HaNikra in the north (Figure 2). It is generally a smooth coastline open to the west that gradually changes in orientation from northeast to almost north, with the exception of Haifa Bay, the Mount Carmel headland and a few small rocky promontories (e.g., Jaffa, Atlit, Akko). The Israeli coast can be divided into four sections: The Southern coast, the Central coast, the Carmel coast, and the Western Galilee coast. In this study we focused on the Central and Carmel coasts.

Along about 45 km from the coast of Israel, a calcareous sandstone cliff (locally known as kurkar) extends parallel to the coastline [35–43]. Tectonically, the coast was generally stable in the Holocene [40,44]. Global sea level curves and geo-archaeological markers along the coast show that since the Last Glacial Maximum, about 20,000 years before present (BP), the sea level rose from about −125 m bsl to the present level and stabilized (±1 m) about 4000 years ago [45–50].

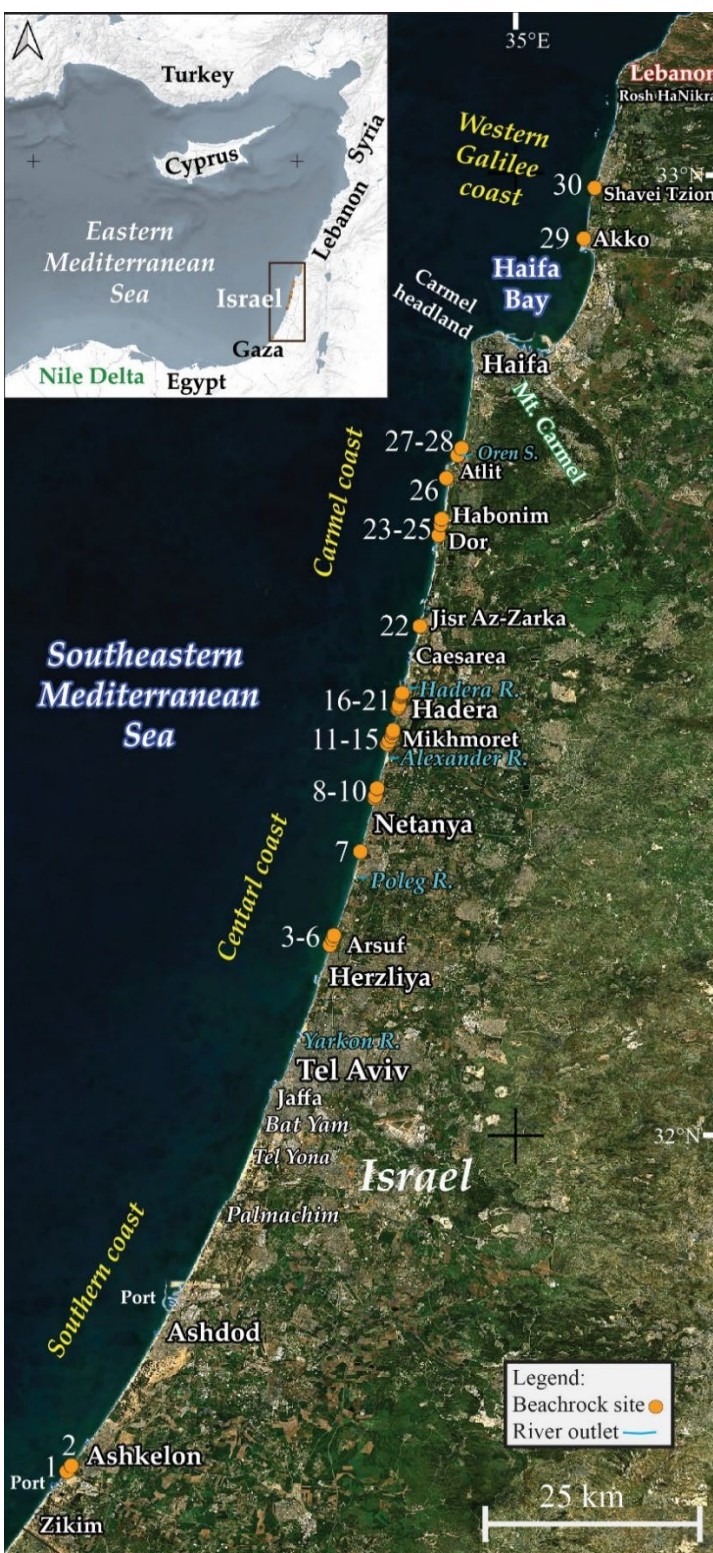

**Figure 2.** The eastern Mediterranean (top left) (Map details: Middle East Mediterranean coast: terrain light background layer by EOX—4326); and the Mediterranean coast of Israel: showing the location of selected BR outcrops discussed in this paper, and relevant geographic features (details: background map: Sentinel-2 cloudless layer for 2020 by EOX– 4326).

The wave climate during the summer (April to October) is characterized by a relatively calm sea with a significant wave height rarely exceeding 2 m (Hs ≤ 2 m). In the winter (November to March) however, cyclonic depressions coming from the south-west/west

(termed Cyprus lows) dominate. The winter wave climate is characterized by alternating periods of calm sea and storm events of up to 5 m Hs [51–53]. Tidal range is very low (0.15–0.40 m), with little or no distinctive influence on sediment transportation [40].

Most beaches are straight and have a moderate slope, are less than 50 m wide and mainly contain fine quartz sand [54]. In some coasts, the beach sediment consists of a mixture of coarse biogenic sand with whole and fragmented shells, pieces of kurkar (sand, gravel, and pebbles) and material from river outlets [55–59].

The coast and its inner shelf, extending from the shore to about 30 m water depth, can be divided into two main sedimentological provinces. The Southern Province, from Zikim to the Akko promontory (northern Haifa Bay), including Southern, Central, Carmel, and Haifa Bay coasts, is considered the north flank of the Nile littoral cell [52,60]. This region mainly consists of fine Nile-derived quartz sand. The Northern Province, i.e., the Western Galilee coast, is a small, isolated, and mainly rocky littoral cell, partly covered with local coarse carbonate sand [55,61]. The fluvial contribution to the coastal sediment is negligible. The rivers bring fine sediment (mostly clay) [57–59], usually during winter storms, which is rapidly washed offshore.

Holocene BR outcrops are described as stair or tile-like laminated blocks, slightly inclined westward towards the sea with the same slope as the hosting beach, and with the same lamination. Beachrock outcrops can stretch hundreds of meters alongshore, being more than 40 m in width, and up to about 1 m in thickness [40,62,63]. Beachrock has a unique and important natural value in Israel. As a hard component placed in an ambient environment, it is a habitat for diverse biota, enriching the regional ecosystem (e.g., [64]). The sedimentological composition of BR is in many cases identical to the four main detrital components composing the non-consolidated coastal sediment in the shoreface: (a) well-sorted mineral and biogenic sand; (b) well-rounded coarse granules of kurkar; (c) complete and broken shells (mainly *Glycymeris*); and (d) other inclusions such as parts of anthropogenic materials and various rocks, gravel, and pebbles (e.g., limestone, chert) [35,65–68].

## 2. Methods

Aiming to characterize the abundance and typical morphology of BR outcrops on the Mediterranean coast of Israel, this study focuses on recording the selected BR outcrop locations, the extent of their lateral exposure, their erosion patterns, and their field relations within the coastal litho-stratigraphic setting. After a preliminary field survey along the Israeli shoreline, 30 sites were chosen for the research. The site spacing was considered an appropriate resolution to evaluate the overall continuity and erosion morphology on the Central and Carmel coasts, with additional study cases at the Southern and Western Galilee coasts (Figure 2). The BR outcrops observed, studied and presented here are only part of the BR outcrop sites surveyed (including Holocene BR along the coastline between Ashkelon and Ashdod, between Ashdod and Palmachim and between Akko and Rosh HaNikra).

The fieldwork included a detailed examination of the outcrops at the selected sites and careful identification of the morphological features of each outcrop. Exact locations were obtained by the GPS navigational system (Google Earth), topographical maps (Survey of Israel, 1:50,000 scale), and aerial images (Survey of Israel, 2021).

A total of 43 beach profiles (lateral transects, perpendicular to shoreline) were recorded using roller tape (measurement gap of 0.2–2.0 m) and a hand-held level. Estimated accuracy of the measured elevations of the beach profile was ±0.3 m. Site elevations were defined using an optic elevation gauge, as well as by manual measurements, all dependent on the nature of the section and the local conditions. The elevations were calculated based on averages of multiple measurements of sea level, and were corrected to the local average tidal range (0.15–0.40 m). Shallow submarine exposures were investigated using scuba or free diving techniques. Given the changes in the sediment cover due to seasonal morphodynamics, and thus the availability of exposed BR outcrops, most of the sites were

repeatedly observed in the spring, at the end of summer, and during winter, to monitor the outcrop exposure with respect to seasonally sediment burial.

Samples were obtained from several locations in each outcrop to validate its definition as BR and to characterize its texture and clastic composition at macroscopic scale [2,69]. Additionally, at Gador I site, drilling was conducted to assess the thickness of the in-situ outcrop and to define the sub-surface beneath it.

## 3. Results: Selected Beachrock Outcrops: Morphological Description

Beachrock outcrops are abundant along most of the Mediterranean coast of Israel, both on sandy and rocky beaches. Overall, 30 sites of BR outcrops are reported here in detail, with particular emphasis on the Sharon coast (between Herzliya and Hadera River) and the Carmel coast (between Hadera River and the Carmel headland) (Figure 2).

Of the 30 sites, two sites are located along the Southern coast of Israel, 19 sites along the Central coast, seven sites along the Carmel coast, and two sites along the Western Galilee coast. Details of the 30 BR sites are shown in Table 1.

**Table 1.** Selected BR outcrop sites along the Mediterranean coast of Israel: Locations and physical properties at transects studied. (Main clastic components: BS: biogenic sand; S: shells; RI and CI: rock, and ceramic inclusions; KG: kurkar granule; QS: quartz sand).

| Site Name (No.) | Coordinates (Lat./Lon.) | Beach Type | Sub-Layer Lithology | Backshore Width (m) | BR Width (m) | BR Height (cm) | BR Main Clastic Components |
|---|---|---|---|---|---|---|---|
| Ashkelon I (1) | 31°39′05″ N 34°32′05″ E | Archaeological Tel | kurkar | 20 | 55 | 30 | QS, S, RI |
| Ashkelon II (2) | 31°39′27″ N 34°32′21″ E | sandy, Archaeological Tel | kurkar | 25 | 22 | 60 | QS, S |
| Arsuf I (3) | 32°11′54″ N 34°48′26″ E | sandy | kurkar | 30 | 14 | 60 | KG, S |
| Arsuf II (4) | 34°48′36″ N 32°12′16″ E | sandy | kurkar | 22 | 23 | 60 | KG, S |
| Arsuf III (5) | 34°48′42″ N 32°12′32″ E | sandy | kurkar | 30 | 17 | 50 | KG, S |
| Arsuf IV (6) | 32°12′29″ N 34°48′41″ E | sandy | kurkar | 30 | 13 | 40 | KG, S |
| Netanya I (7) | 32°17′43″ N 34°50′20″ E | sandy | kurkar | 20 | 12 | 20 | KG, S |
| Netanya II (8) | 32°21′03″ N 34°51′14″ E | sandy | kurkar | 28 | 14 | 25 | KG, S |
| Netanya III (9) | 32°21′12″ N 34°51′16″ E | sandy | kurkar | 34 | 15 | 70 | KG, S |
| Netanya IV (10) | 32°21′39″ N 34°51′22″ E | sandy | kurkar | 30 | 30 | 30 | KG, S |
| Mikhmoret I (11) | 32°24′25″ N 34°52′01″ E | sandy | kurkar | 25 | 12 | 30 | KG, S |
| Mikhmoret II (12) | 32°24′36″ N 34°52′07″ E | sandy | kurkar | 40–44 | 10–12 | 15–25 | KG, S |
| Gador I (13) | 32°24′55″ N 34°52′16″ E | sandy | kurkar | 63 | 10 | 80 | KG, S |
| Gador II (14) | 32°25′09″ N 34°52′19″ E | sandy | kurkar | 60 | 17 | 68 | KG, S |
| Gador III (15) | 32°25′16″ N 34°52′22″ E | sandy | kurkar | 42 | 10 | 55 | KG, S |
| Binyamin Cove I (16) | 32°26′36″ N 34°52′39″ E | sandy-pebbles | kurkar | 12–17 | 8–10 | 20 | KG, S |
| Binyamin Cove II (17) | 32°26′42″ N 34°52′42″ E | sandy | kurkar | 2 | 9 | 30 | KG, S |
| Angler Cove (18) | 32°26′51″ N 34°52′44″ E | sandy | kurkar | 44–58 | 40 | 20–54 | KG, S |
| Hadera River I (19) | 32°27′24″ N 34°52′51″ E | sandy | kurkar | 68 | 30 | 66 | KG, S |

**Table 1.** *Cont.*

| Site Name (No.) | Coordinates (Lat./Lon.) | Beach Type | Sub-Layer Lithology | Backshore Width (m) | BR Width (m) | BR Height (cm) | BR Main Clastic Components |
|---|---|---|---|---|---|---|---|
| Hadera River II (20) | 32°27′30″ N 34°52′54″ E | sandy | kurkar | 70 | 20 | 12 | KG, S |
| Hadera River III (21) | 32°27′36″ N 34°52′56″ E | sandy | kurkar | 50 | 11 | 40 | KG, S |
| Jisr az-Zarka (22) | 32°31′45″ N 34°54′03″ E | sandy | Kurkar | 40 | 30 | 30 | KG |
| Habonim—Dor I (23) | 32°37′26″ N 34°55′11″ E | rocky | kurkar | 55 | 70 | 80 | S, RI |
| Habonim—Dor II (24) | 32°38′07″ N 34°55′19″ E | rocky | kurkar | 12 | 2 | 20 | S |
| Habonim—Dor III (25) | 32°38′27″ N 34°55′23″ E | rocky | kurkar | 20 | 6 | 20 | S |
| Atlit I (26) | 32°40′59″ N 34°55′41″ E | rocky | kurkar | 65 | 20 | 80 | BS |
| Atlit II (27) | 32°42′25″ N 34°56′23″ E | sandy | clay | 35 | 10 | 20 | BS, CI |
| Atlit III (28) | 32°42′51″ N 34°56′37″ E | sandy | clay | 40 | 4 | 20 | BS |
| Akko North millstone quarry (29) | 32°55′55″ N 35°04′14″ E | rocky | kurkar | 30 | 28 | 80 | BS, CI |
| Shavei Tzion Beit HaEmek Stream outlet (30) | 32°59′05″ N 35°04′54″ E | rocky | kurkar | 60 | 60 | 100 | BS, S, RI |

*3.1. Southern Coast*

The Southern coast of Israel extends for about 51 km from Zikim to Tel Yona (Figure 2). This section includes wide (50–100 m) and moderate sandy beaches backed by low dunes or kurkar hills (e.g., Zikim, south Ashdod), and narrow beaches bounded by a steep and unstable kurkar coastal cliff (e.g., north Ashkelon and Yavne Yam).

Beachrock exposures are abundant along the coastal segment between Ashkelon Port and Tel Ashkelon beach (located about 3 km to the north) and along most of the beaches from north Ashkelon to Ashdod, and from north Ashdod to Yavne Yam.

Two sites were chosen from the Southern coast as study cases of BR outcrops, as they are located closest to the main sand source of the beach (i.e., the Nile Delta).

3.1.1. Ashkelon

The backshore near the sites studied (Ashkelon I, II; Table 1: 1, 2) is a coastal kurkar escarpment rising to 18 m asl (in places with eroded extensions towards the sea) and the remains of Tel Ashkelon's eastern margins [40]. The beach is narrow (5–20 m), its sediment consists mainly of quartz sand and shells, kurkar particles, and presence of archaeological debris. Most of these outcrops are covered by unconsolidated sediment or exposed.

At site Ashkelon I, (from Ashkelon Port to Tel Ashkelon), the backshore consists of a low (2–3 m) kurkar terrace. BR is exposed (Figure 3) as a comprehensive (about 55 m) band, extending from the shoreline down to about 3.5 m water depth. This site morphology is characterized by a wide (20–25 m) deposit of in-situ BR plates facing the sea. At the east in-situ BR band margin, there is a stair-like bump on the BR surface, created by erosion and scouring at the sea side of the BR deposit. Many unattached BR plates (up to 0.3 m high and 2.5 m long) are scattered along the water line and backshore. The submerged part consists of a large BR plates (up to 0.3 m high and 4 m long) apron and thinner BR plates, concentrated at the apron's edge, at a depth of about 3 m.

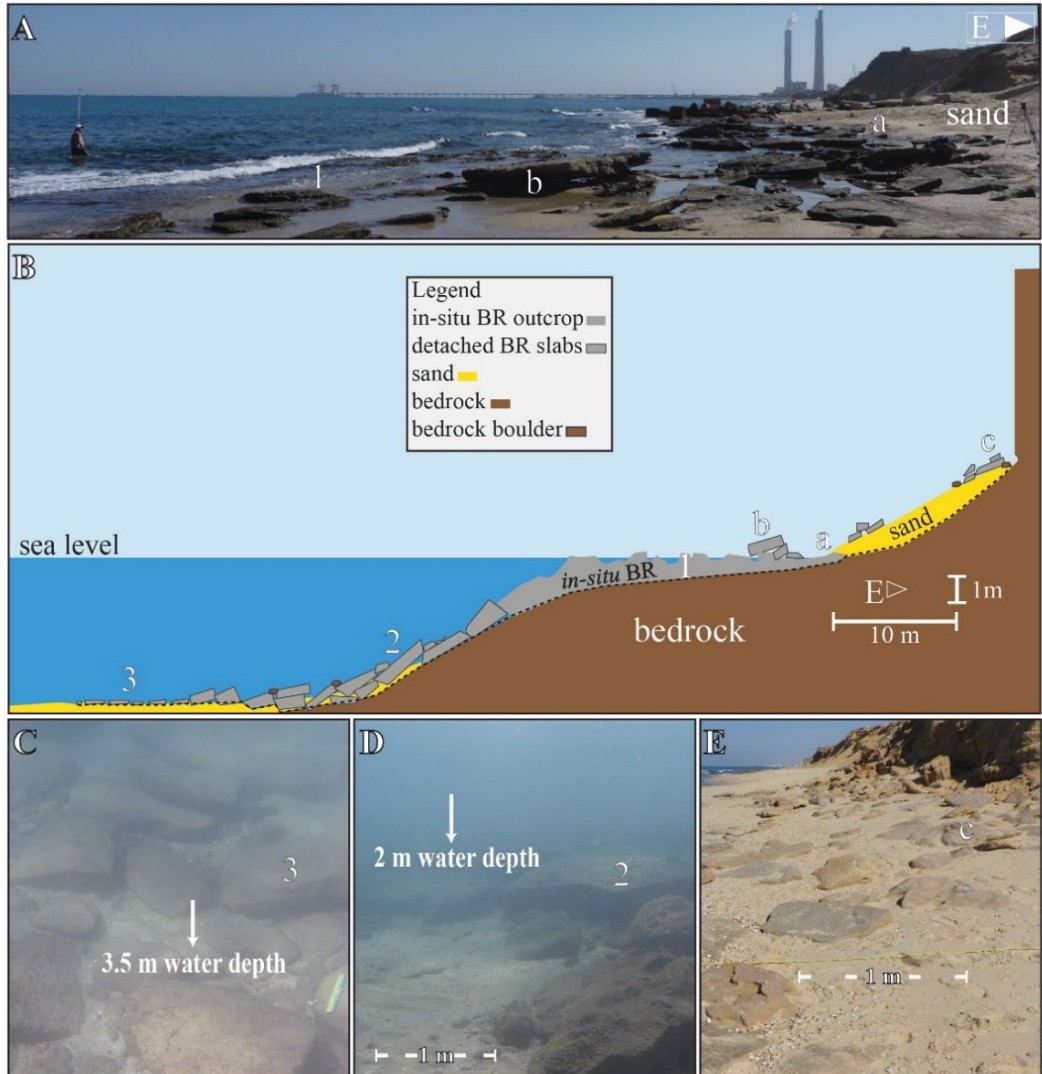

**Figure 3.** Beachrock outcrop south of Tel Ashkelon: (**A**) view of the site (**1**: in-situ BR outcrop, (**a**) BR: unconsolidated sediment contact, (**b**) clast of BR slabs; (**B**) schematic cross-section of the outcrop (VE = 4, **2**: submerged BR slab apron (see (**D**)), **3**: clast of BR plates (see (**C**)), (**c**) clast of BR plates backshore (see (**E**)), dotted line marks uncertain depth of unit contact depth, other notes are as in (**A**); (**C**) submerged clast of BR plates at the sea side edge of the outcrop (scale at bottom right is 0.3 m); (**D**) submerged BR slab apron (looking north); (**E**) beachrock plates at backshore (looking north).

Ashkelon II site is located on the coast of Tel Ashkelon, near the 'Pillar Wall', a remnant of the Crusader coastal fortification built at the beginning of the 2nd millennium [40]. The land side face of the intertidal BR band is a low (~0.2 m) stair. Detached blocks (up to 2 m long) lie east of it in a shallow trough. A few stair-like bumps extend along the BR band surface. At this site, a wide (30 m) band of high (up to 0.7 m) BR slabs are found at 3 m depth, 50–60 m off the shoreline. These slabs probably settled towards the hard bottom due to erosion and scouring of the underlying loose sediment. Unconsolidated sediment covers the area between the two exposures (BR band and intertidal BR band).

### 3.2. Central Coast

The Central coast of Israel extends for about 54 km from Tel Yona to Hadera River outlet. The main part of this section is extending from the Yarkon River outlet to the Hadera River outlet (Figure 2) and known as the Sharon coast.

Along most of the Sharon coast rises a high (up to 50 m), steep and unstable cliff, and at its foot stretches a narrow sandy beaches. Relatively wide natural beaches in central Israel are found between Tel Yona and the south of Bat Yam, as well as in places where the continuity of the coastal cliff has been interrupted due to streams that cross it and flow into the Mediterranean (e.g., the Yarkon, Poleg and Alexander rivers) (Figure 2).

Beachrock outcrops are abundant along most parts of the Central coast, although large sections have been demolished or covered with sand due to urban development, especially along the Tel Aviv coast.

### 3.2.1. Southern Sharon Segment

This coastal segment extends from north Herzliya (Apollonia) to the Poleg River outlet. It's characterized by a high coastal cliff and narrow (less than 25 m) beaches, consisting of fine to coarse sand, kurkar fragments, broken shells, and detrital quartz grains with coarse gravel (reworked laminar kurkar fragments).

The in-situ BR band (Table 1: 3–6) is almost continuously exposed from Apollonia to Poleg River (with gaps 100–200 m long). The typical (Table 1) in-situ BR outcrop appearance is represented by the exposures observed along this section of the study area.

The outcrop's upper surface (14–23 m wide) dips slightly seawards, and on its landward edge, a micro-cliff, e.g., [70] is formed (up to ~ 0.5 m high). In most cases, exposed detached plates are situated at the foot of the micro-cliff and up to a few meters from it. Detached BR plates lie at the sea side of the in-situ beds, dipping more steeply seaward than the in-situ outcrop's surface. Near the submerged sea side face of the outcrop, some plates are missing and reveal the sandy bottom. Beachrock boulders, parts of the former in-situ BR, are situated close to both the sea and land sides of the main in-situ deposit (the number of unattached plates on the sea bottom and on the coast varies between the sites and changes throughout the year). The north and south edges of the outcrops often form a steep stair, with plates lying nearby. The chosen sites demonstrate the variance in beach profile. For instance: Arsuf I (Figure 4A,B) shows an abraded surface, many detached BR plates in its margins, and most of the beach (backshore width is up to 14 m). Arsuf II (Figure 4C,D) and Arsuf III, a wide (about 20 m) BR exposure, create narrow (up to 1 m wide), sandy troughs east of the stair-like micro-cliff. Part of the micro-cliff here borders an elongated kurkar platform (a relic of a coastal cliff collapse) and long detached blocks of BR (up to 3 m) dropped from the land side micro-cliff and settled beside it. In the BR outcrop at Arsuf IV, the BR surface is badly abraded, many detached BR blocks are present inside the trough, and many kurkar boulders are scattered along the backshore and on the BR.

### 3.2.2. Netanya Segment

This segment extends 15 km from the Poleg River outlet to Alexander River Outlet. The cliff escarpment along this segment is subjected to severe retreat, and some engineering solutions have been carried out in places at the base of the cliff to reduce its collapse.

At site I, Argaman Beach (Table 1: 7), the beach is narrow, the loose sediment is coarse and includes many kurkar pebbles. Beachrock outcrops along the water line exhibit typical morphology. It is about 12 m wide, and its land side front is ~0.2 m high.

At Sironit Beach, about 4 km north of Argaman Beach, two detached breakwaters protect a wide beach, with public installations and various structures at the base of the cliff. Beach profile measurements at three sites (Table 1: 8–10) present a variation in the nature of the profile in each site. Netanya site II, Havatzelet Hasharon site (Figure 5A,B) is part of the beach under construction at the backshore, aiming to prevent cliff retreat. The loose sediment consists mostly of quartz sand. Flat surfaces of BR beds are exposed at the intertidal zone, projecting up to 0.2 m from the loose sediment profile. The rest of the BR outcrop is covered. At Netanya site III, Shoshanat Ha'amakim (Figure 5C,D), part of the BR band is broken by a collapse of boulders up to 2.5 m long. The BR is covered by a platform (several cm) of biogenic growth in some parts. At Netanya site IV, Neurim site

(Figure 5E,F), the BR section meets an elongated kurkar block at its land-facing front. The combined structure creates a sediment spit abutting the micro-cliff.

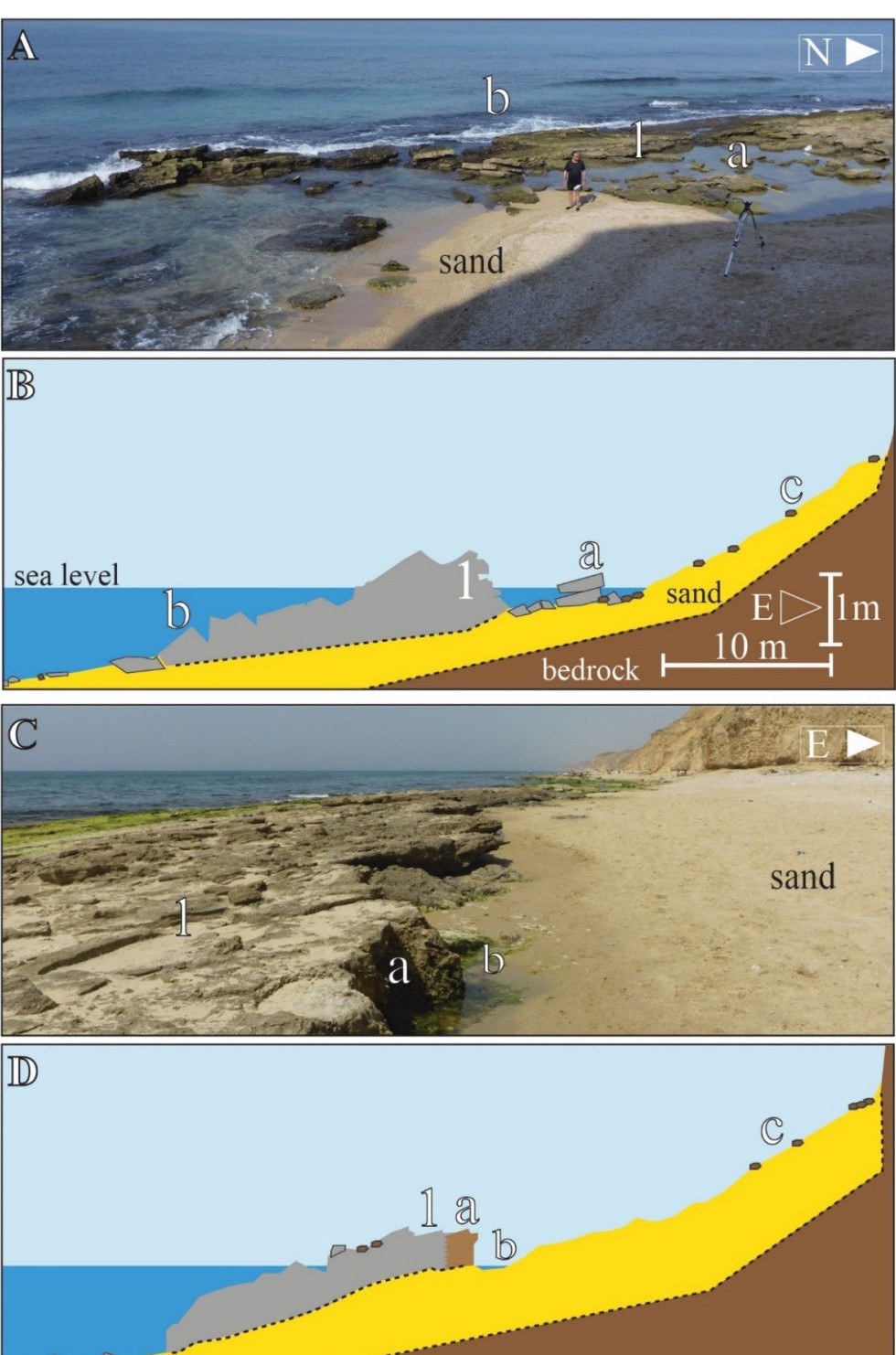

**Figure 4.** Beachrock of the southern Sharon beaches: (**A**) view of the Arsuf I site (**1**: in-situ BR outcrop, (**a**) clast of BR boulders, (**b**) sea side ledge of the BR exposure; (**B**) schematic cross-section of the outcrop (VE = 4, (**c**) kurkar boulder backshore, other notes are as in (**A**), dotted lines mark uncertain depth of unit contacts); (**C**) view of Arsuf II site (**1**: in-situ BR outcrop, (**a**) elongated kurkar platform bordering the BR outcrop at land side front, (**b**) shallow trough); (**D**) schematic cross-section of the site (VE = 4, (**c**) kurkar boulder backshore, other notes are as in (**C**), dotted lines mark uncertain depth of unit contacts).

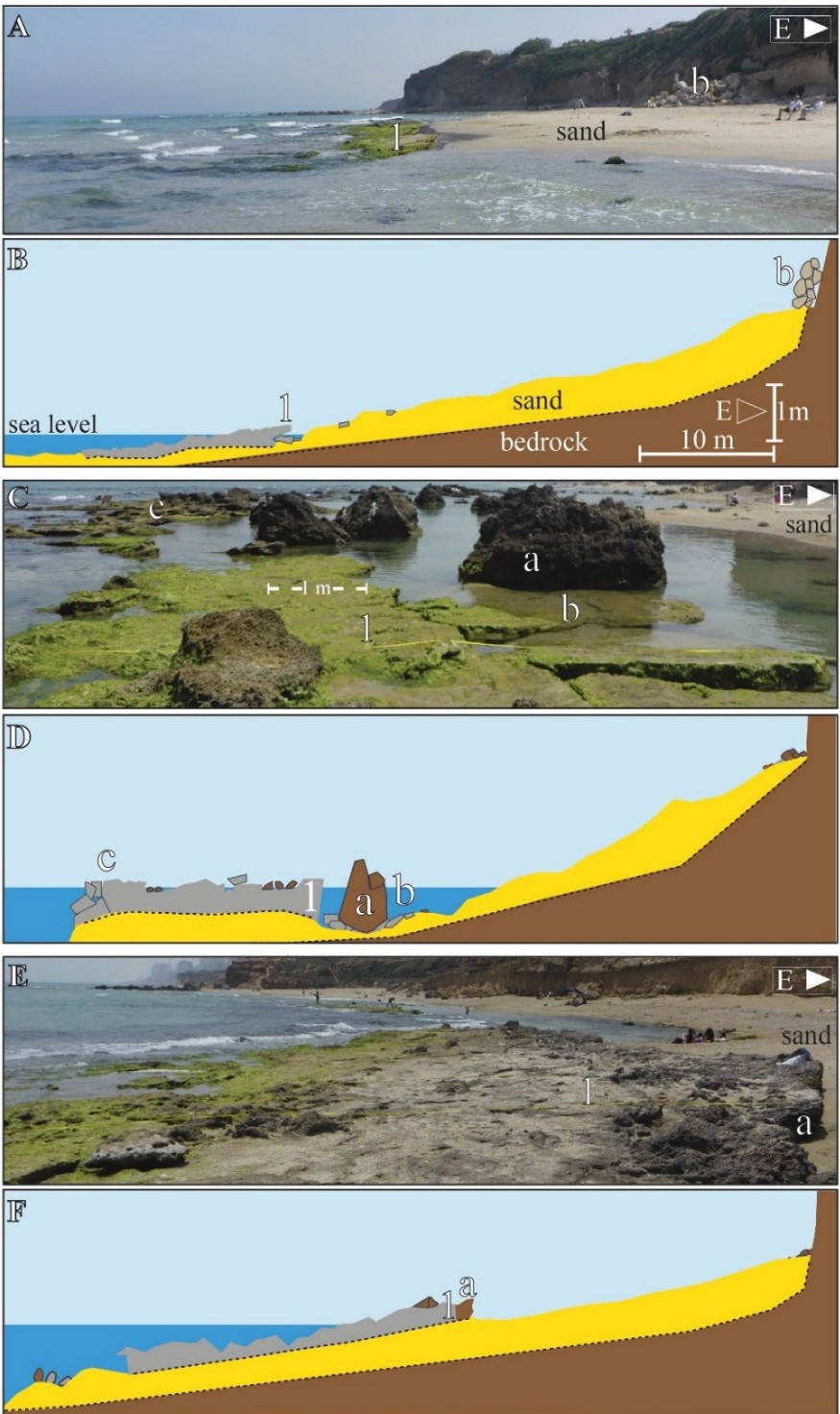

**Figure 5.** Beachrock exposures north of Netanya and south of Alexander River outlet: (**A**) view of Havatzelet Hasharon (Netanya site II) (**1**: BR outcrop, (**b**) artificial ramp); (**B**) schematic cross-section of the site (VE = 4, notes are as in (**A**), dotted lines mark uncertain depth of unit contacts); (**C**) view of Shoshanat Ha'amakim (Netanya site III) (**1**: BR outcrop land side front, (**a**) kurkar boulder from cliff collapse, (**b**) beachrock slabs under boulder, (**c**) clast of broken BR boulders); (**D**) schematic cross-section of the site (VE = 4, notes are as in (**C**), dotted lines mark uncertain depth of unit contacts); (**E**) view of Neurim site (**1**: BR outcrop, (**a**) elongated friable sandstone front bordering the BR outcrop at land side); (**F**) schematic cross-section of the site (VE = 4, notes are as in (**E**), dotted lines mark uncertain depth of unit contacts).

### 3.2.3. Mikhmoret Segment

This segment can be divided into three sectors: (1) Alexander River outlet beach—a wide beach backed by dunes. The beach contains mainly loose sand, granules and pebbles, which lying on top of a Pleistocene paleo-embayment; (2) Tel Mikhmoret area—a rocky coast with a steep escarpment (up to 5 m) and significant collapse of kurkar blocks. The coast north of Tel Mikhmoret is rocky, and includes two sheltered coves: the southern cove, is a small anchorage for boats, and the northern cove is a public swimming beach; (3) The northern sector (about 2 km long), where the present study sites are located, is part of the Gador Nature Reserve, and is considered a 'natural beach' with no significant impact of marine construction on the local coastal morphology. This sector is characterized by natural evolution, effected mainly by the retreat of the cliff front. Many large (up to a few meters high) calcarenite ('Tel Aviv kurkar') boulders are present along the cliff base. The coastal cliff in this segment comprises a series of long (300–800 m) concave arches. A few natural channels drain its hinterland. Most of the beach is sandy, with a significant presence of kurkar and BR pebbles and boulders. Intertidal sediment mainly consists of kurkar fragments (granule–sand size), shells (whole, crushed, and fragmented), and quartz sand. Along this segment, the BR outcrop appears as a semi-continuous band (Table 1: 12–15), bordered on its and land side parts by unconsolidated sediment. In the Gador Nature Reserve, a shallow trough (or lagoon) borders the BR band and the sandy profile most of the year.

No BR is exposed along the shoreline at Mikhmoret I site ('Banana Beach', Table 1: 11). However, a submerged BR band is exposed about 100 m from the waterline, at about 2 m depth, close to a longitudinal kurkar reef. Around Mikhmoret II site ('Kushi Beach'), in-situ BR outcrops are exposed near the water line. Part of this outcrop lies over an elongated kurkar platform. The BR band at Gador I site (Figure 6, Table 1: 13) was chosen as a case study, and punched using a pneumatic cup (0.5 m in diameter) drill down to 1 m depth (to assess the thickness of the in-situ outcrop and to define the sub-surface beneath it), showing that its in-situ crest is 0.8 m thick and that it is situated above sand. The sea side part of the BR band is submerged under about 1 m of water, and some rounded boulders lie on the seafloor over it. About 60 m seawards of the main in-situ BR band margin, a BR parallel to shore exposure was observed. These long (more than 50 m) and wide (up to 20 m) band of coarse-grained BR plates are exposed at a water depth of 2.0–2.5 m.

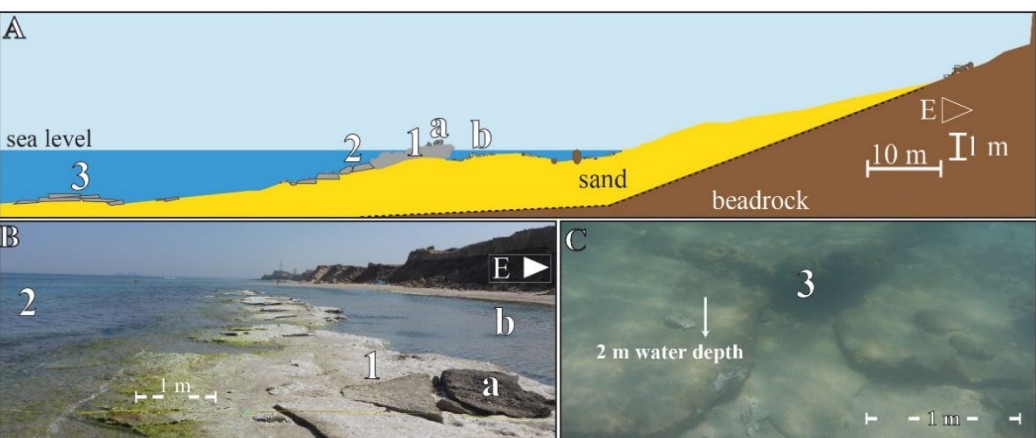

**Figure 6.** Beachrock exposure at Gador I site (north of Mikhmoret): (**A**) schematic cross-section of the site (VE = 4, **1**: in-situ BR outcrop, **2**: sea side part of the BR band on the shore face; **3**: submerged clasts of wide BR slabs at about 2.5 m water depth (which underwent settling), (**a**) clasts of BR plates on top of the in-situ outcrop, (**b**) clasts of BR boulders at the trough, dotted line marks uncertain unit contact depth); (**B**) view of the site (notes are the same as in (**A**); (**C**) closeup (looking west) on the submerged BR section, loose blocks up to 3 m long, which underwent settling.

The BR exposure at Gador II is a flat platform. The width of the main part is about 20 m and the overall exposed band, including detached BR plates settled at its margins, is about 40 m. The micro-cliff is ~0.3 m high, projecting over a notch-like formation. A sand spit is present in front of the outcrop, and the trough is relatively narrow (about 2 m).

The BR exposure at Gador III site is located mainly at water level. Its upper bed surface is flat and smooth. Its land side front is a shelf-like micro-cliff. A trough separates the BR band and the backshore during the winter, and angular boulders of kurkar and BR are scattered on its bottom. During the summer, parts of these outcrops are covered by sediment.

### 3.2.4. Hadera Segment

This segment is characterized by differentiation in coastal cliff continuity. Around Binyamin Cove and Angler Cove sites, the cliff appears as a series of humped-back hills. Near the Hadera River outlet, the sandy beach is backed by dunes, and only low (up to 5 m) kurkar bluffs are present. The Olga-Binyamin Cove area was affected by major cliff collapse events in recent years, and is also subjected to modern anthropogenic development, which has disturbed its natural appearance [68]. BR appear in the north and south extremities of the cove as a band (Table 1: 16, 17) and between and under collapsed kurkar boulders (up to 4 m high).

At Binyamin Cove I site, BR exposures are a narrow band of beds of typical morphology up to 0.2 m in thickness and 10 m in width. This BR outcrop continuation to the north is located between and under a kurkar collapse of large boulders up to 4 m high, next to a loose sediment granule–pebble size mixture. BR beds and BR remains are found up to 30 m from the cliff. Binyamin Cove II site is in the northern part of the cove, located south of a cliff collapse, where a modern wall was constructed on the water line and an artificial terrace backshore. The loose sediment is a coarse granule–pebble mixture. A narrow (up to 10 m) BR and low (about 0.2 m) band of typical morphology is exposed, partly broken by collapsed kurkar boulders. Some detached BR plates are found near the outcrop margins, and several kurkar boulders are cemented to the upper BR bed surface.

At the Angler Cove site (Table 1: 18), the BR outcrop is about 200 m long, and about 30 m wide. Backshore, the unconsolidated sediment profile is close to a disturbed hill. Beach sediment is sand–gravel, with many kurkar and BR boulders along the shoreline and backshore. The BR exposure here is characterized by its various erosion surface features, bedding patterns, internal texture, and consolidation level. The western part of the exposed upper surface is abraded and eroded (including bumps, depressions, pits, holes, various patterns of cracks, dislocation slots, a stair-like micro-cliff, elongated protrusions, channels, and bores). Kurkar and BR boulders were found re-cemented in places to the outcrop's upper surfaces. The eastern edge of the outcrop borders and covers a loose beach sediment deposit. The exposed BR beds near the unconsolidated sediment–BR contact zone are characterized by relatively smooth surfaces.

The BR band is intermittently exposed between Angler Cove and Hadera River outlet (Table 1: 19–21) along the wide flat sandy beach. At the Hadera River I site the BR exposure is characterized by high fronts (up to ~ 0.65 m) at the band's sea and land side fronts. On the sea side front, detached BR plates are present at the contact point with the covering sediment. The micro-cliff faces a trough (~ 0.5 m deep), and its surface includes shelf-like bulges. At the Hadera River II site (Table 1: 20), a sand spit is close to the BR outcrop land side front. At the Hadera River III site (Table 1: 21), the BR band is exposed (11 m wide), located about 30 m west of the waterline, situated entirely in or under the water level.

### 3.3. Carmel Coast

The Carmel coast extends for about 42 km from Hadera River outlet to the Carmel headland. This section can be sub-divided to two parts: The Southern Carmel coast (between Hadera River outlet and Atlit), and the Northern Carmel coast (between Atlit and the Carmel headland). The Carmel coast includes different types of sandy beaches and

rocky coasts, and is characterized by kurkar ridges, running parallel to the coastline, and between them, troughs filled with clays, sands, and alluvial sediments. East-west oriented channels of ephemeral streams drain the western slopes of Mount Carmel, crossing the coastal plain on their way to the Mediterranean [45].

### 3.3.1. Jisr az-Zarka

Jisr az-Zarka coast is located north of Caesarea (Figure 2), and is characterized by kurkar shoals, which are preserved at different elevations. The coast is formed of sections moderately arched concave seawards, up to 1 km long and up to 90 m wide. The hinterland consists of sand 0.5–6.0 m thick. South of the Jisr public swimming beach, a wide BR band about 1.5 km long outcrops intermittently along the curved coast. The backshore is formed chiefly of sands covering low kurkar hill topography. The kurkar shows clear bedded structures exposed near the shoreline in the form of serrated surfaces. The beach profile is formed of sand and shells, and many BR and kurkar boulders are present along the backshore. At the Jisr az-Zarka site (Table 1: 22) the BR band is exposed along the intertidal zone and is covered by sediment on its sea side margin. At the land side margin, a low (up to 0.3 m) micro-cliff is present, approaching the kurkar slope, or bordered by the covering sediment of the beach profile. The BR outcrop's upper surface is flat, and includes various erosion morphological features. Further south, wider (about 20 m) BR exposure was observed, including a thicker micro-cliff and concentrations of archaeological ceramic sherds in specific locations. Little or no BR exposures are found north from Jisr or along the wide sandy beaches of the southern Carmel coast up to Tel Dor.

### 3.3.2. Habonim and Dor North

The Dor-Habonim segment is a mostly rocky beach with several sandy pocket beaches. Its coastline includes about 30–35 small (up to about 300 m wide) coves, with kurkar spurs and capes between them. Various rocky morphological formations are present, such as vertical rocks, peg-like coves, notches, abrasion platforms, grottos, crannies, arches, caves, niches, and chimneys.

The beach sediment consists of sand and shells, but in some places large parts of the beach profile consist only of shells. While in the northern part of Habonim beach and towards Atlit, the unconsolidated sediment beach profile is thick and no BR is usually exposed, around the rocky coast of Habonim Nature Reserve Park, a few BR outcrops are continuously exposed.

The Habonim—Dor site I (Beachrock Cove Dor north site, Table 1: 23) is located north of Tel Dor and its coves. This BR outcrop is long (more than 150 m), wide (more than 60 m), and thick (beds up to 0.8 m), creating a complete beach profile that consists of BR beds and piles of BR blocks (Figure 7). Beach sediments comprised of shells are concentrated on the back edge. The western marginal beds are submerged and slope more steeply seawards than the rest of the outcrop's surface. The slope of BR plates extends for about 20 m seawards, up to about 2 m water depth. Over the main in-situ BR band, smaller BR plates lie on the seafloor for about 20 m, up to a depth of about 4 m, and at 40 m from the water line (about 100 m from the backshore). From there BR is replaced by kurkar pebbles and sand.

At Habonim—Dor site III, a 200 m long, thin (0.15–0.30 m) BR bed, containing many shells (such as the loose sediment in the beach profile) outcrops as a narrow band (about 5 m) on the water line (Figure 8, Table 1: 25). The BR beds are partly covered with sediments. BR blocks and pebbles are concentrated in places along the shoreline and backshore. At Habonim—Dor site II (Table 1: 24), a similar configuration, although much smaller, is found about 500 m south, in a wide (about 35 m) niche at the kurkar, forming a small beach on the abrasion platform. A micro-cliff occurs near the shoreline, as well as the BR-kurkar contact point. Part of the outcrop is covered by sand and shells. Little BR is exposed north of Habonim—Dor site III, over the Habonim National Park beach, and along the wide sandy

beaches of Neve Yam and Tel Nami. In places, BR surfaces, cut by furrows, are exposed at the intertidal zone above clay sub-layers.

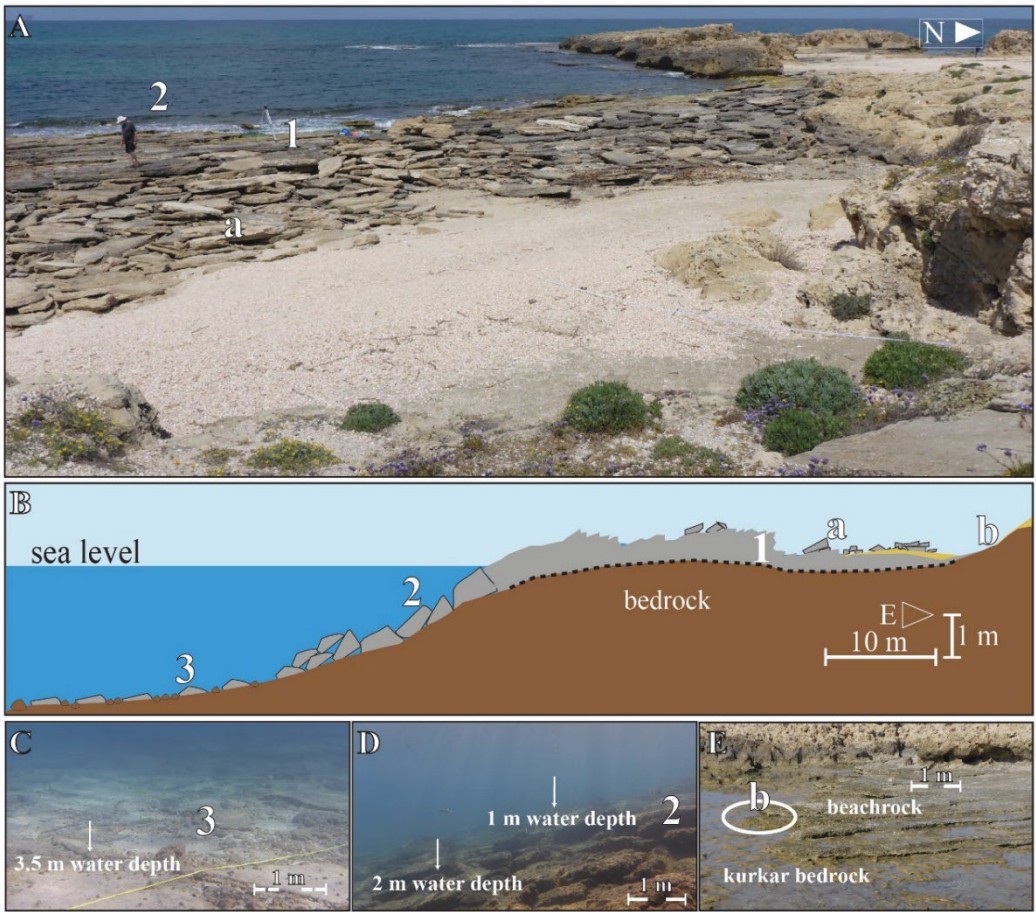

**Figure 7.** Beachrock exposure at Beachrock Cove site, north of Dor: (**A**) view of the site (**1**: BR outcrop; **2**: broken BR slabs at sea side, (**a**) BR slabs scattered on shore); (**B**) schematic cross-section of the site (VE = 4, **3**: BR plates paving seafloor at about 3.5 water depth, (**b**) BR: bedrock contact backshore, dotted line marks uncertain unit contact depth, other notes are as in (**A**)); (**C**) close-up (looking south-west) of BR exposure sea side edge (**3**: as in (**B**)); (**D**) close-up (looking north) of BR exposure sea side of the BR outcrop (**2**: as in (**A**,**B**)); (**E**) close-up (looking south-west) of BR–bedrock contact backshore ((**b**) as in (**B**)).

### 3.3.3. Atlit

(a)   Atlit South

The Atlit I site (Nun Cove) is situated south of Atlit Bay. The BR outcrop (Table 1: 26) has a stair-like thick landward micro-cliff of 0.8 m above sea level, lying on the kurkar slope. BR boulders are piled in a few areas along the beach profile. This place is highly disturbed by the influence of sea walls and demolition by heavy machinery in the area, on its southern part, where the channel crosses through the outcrop.

(b)   Atlit North

The beach of Atlit Bay (50–150 m wide) separates the Bay and Dor trough (situated between the Dor coastal kurkar ridge and the Atlit ridge) in the south and between the Bay and the Megadim trough (situated between the Megadim and Atlit kurkar ridges) on the east. The narrow beach profile (Figure 9) in the Atlit II site is bordered at the backshore by a wall, supporting the terrace of a Crusader cemetery [40]. Toward the south, the beach has been altered by construction. The steep beach profile consists of loose, fine, well-sorted, crushed bioclasts and quartz sand. Storms revealed the BR (Table 1: 27) outcrop (up to

0.2 m thick, of moderate–weak cementation level bed), lying on a dark clay layer [71] and characterized by its surface micro-topography. Furrows cut the BR outcrop surface here. Many fragments of ceramic vessels were found inside BR in this site, including part of a ceramic cone-shaped jar dated to the 7th–8th centuries BC [72] (p. 492) cemented inside the BR bed. The BR band is situated in the upper intertidal zone at about 10–15 m from the shoreline. Detached blocks are piled on top of the outcrop and backshore.

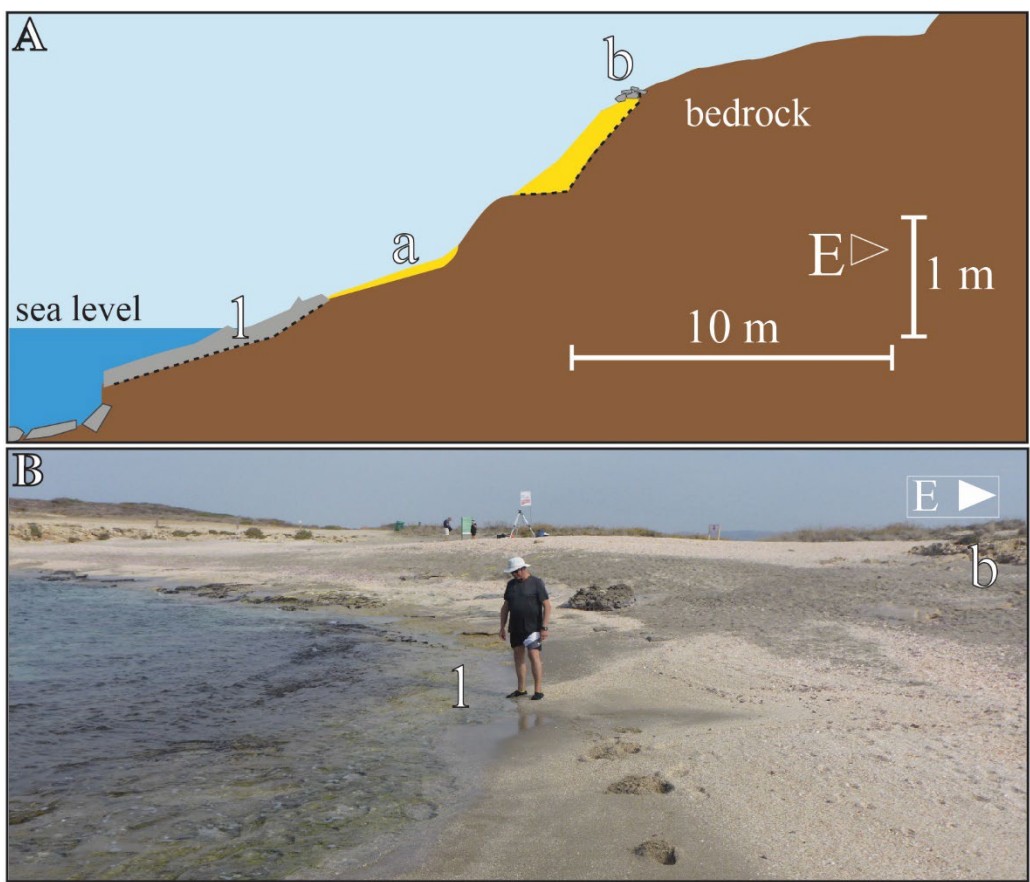

**Figure 8.** Beachrock exposure in Habonim (Bay 4). (**A**) schematic cross-section of the site (VE = 4, **1**: BR outcrop, (**a**) beach surface sediment consisting of sand and shells, (**b**) beachrock boulders concreated at backshore, dotted line marks uncertain unit contact depth); (**B**) view of the site (notes are as in (**A**)).

North of the Oren stream outlet, the beach profile consists of loose, fine, well-sorted, crushed bioclasts, and quartz sand. The BR exposed at the Atlit III Beit Nili site (Table 1: 28) is a narrow layer, only a few meters wide, and about 0.2 m thick. The BR band is situated in the upper intertidal zone, located at about 10–15 m landwards from the shoreline. Detached blocks are piled around. The BR outcrop lies directly on a dark clay unit. Plates of dark BR beds were located under the water at a depth of about 1 m, in front of the west front of the BR. This site is covered (up to 0.5 m) by unconsolidated sediment, only emerging intermittently during the year, and is exposed after extremely stormy weather during the winter.

### 3.4. Western Galilee Coast

The Galilee coast, extending about 20 km from Akko to Rosh HaNikra, is part of an local littoral cell, with a distinct sedimentary regime. Current intensity and direction change frequently, mainly due to the shoreline direction (North-West, azimuth 12–10°, almost parallel to the approaching wave fronts), and rugged character (including islands, reefs, abrasion platforms, submerged canyons, and submerged kurkar ridges) [40]. Beachrock

exposures are abundant along most of this coastal segment. Two sites were chosen as study cases of BR outcrops in this environment.

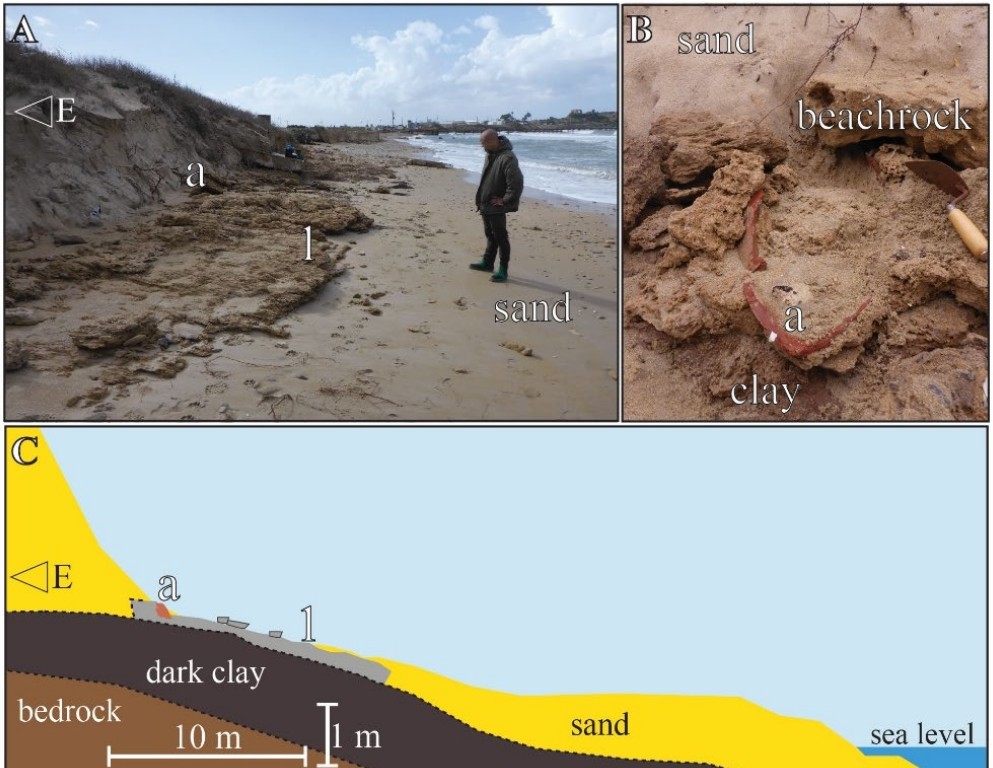

**Figure 9.** Beachrock exposure near the Crusader cemetery at Atlit: (**A**) view of the site (**1**: BR outcrop, (**a**) ceramic jar consolidated inside the BR); (**B**) close-up (looking east) of outcrop surface ((**a**) as in (**A**)); (**C**) schematic cross-section of the site (VE = 4, dotted line marks uncertain unit contact depth, other notes are as in (**A**)).

### 3.4.1. Western Galilee Site I—Akko North Millstone Quarry

The site is located at the rocky beach between Akko promontory and Bustan HaGalil beach. At the Akko site (Table 1: 29) the narrow, steep beach sediment profile is bordered backshore by a modern sea wall and stairs. Sediment is composed of coarse sand and shell fragments. The BR outcrop is wide and thick, and the in-situ part is mainly located on kurkar bedrock. At this site, a millstone quarry was discovered during a survey of the shoreline conducted in 1998 [73]. Dozens of rounded rock cuts, negatives of detached stones, and stones that broke during the process remain in-situ. This quarry also includes straight tunnels and stairs. In vicinity, a pile of kurkar ashlar stones was found consolidated inside the BR surface. Nearby is a concentration of ceramic sherds consolidated inside BR beds; this assemblage was suggested to be Roman–Byzantine. The seaward facing margin of the BR outcrop borders a biogenic build-up, most likely growing over the collapse of BR beds. The BR outcrop's relatively rough upper surface is characterized by troughs (some filled with seawater) and stair-like micro-cliff morphology. The backshore facing front is high (about 0.5 m), forming a straight or shelf-like micro-cliff. Detached BR blocks lie at the foot of the micro-cliff. Over it is a wide trough containing sand and gravel–pebble size sediment.

### 3.4.2. Western Galilee Site II Shavei Tzion (Beit HaEmek Stream Outlet)

The sampled site (Table 1: 30) is located at the Beit HaEmek stream outlet. The BR outcrop at this site is part of a long (about 800 m) BR band that extends from about 100 m north of the stream outlet and about 700 m south of a modern breakwater (built in 1950 and lengthened in 1974). The kurkar ridge at the backshore is very low and is not a distinctive

topographic feature. The beach profile consists of a moderate kurkar slope with BR beds on top. Loose sediment is concentrated mainly backshore. Many BR plates were used for constructing the breakwater and the path at the backshore. The BR outcrop is long (more than 100 m) and wide (more than 60 m). A lateral section of the beach profile was measured along a wide exposed part of the outcrop, representing its main characteristics: detached BR boulders lying on upper BR in-situ beds, BR boulder heaps backshore, a series of troughs and stair-like bump lines, BR blocks dipping seawards at the outcrop's front facing the inshore, a high micro-cliff, and relatively wide flat surfaces, especially at the landward extremities. Submerged BR beds (up to 1 m high) extending up to about 1 m water depth, replaced by detached BR beds, form a steep apron extending seawards up to about 2.5 m water depth, bordering the seafloor, about 5 m from the shoreline.

*3.5. Summary*

Beachrock outcrops morphological features were observed along a significant part of the Israeli shoreline, including representation of various coastal constellations (e.g., sandy, rocky, and archaeological Tel). On the Sharon coast, BR outcrops are exposed on both sand and gravel beaches (10–50 m wide), mostly backed by a coastal cliff (Arsuf, Netanya, Mikhmoret II, and Gador). Mikhmoret I, Binyamin Cove, and Angler Cove sites are located inside coves. Other sites are in a wide sandy beach with sand dune backshore (Hadera River, Table 1:19–21). At the Carmel coast, BR outcrops were observed both in the rocky, part of the coastline (Habonim–Dor, Atlit I), where the unconsolidated sediment coverage is thin or totally absent, and along the sandy beach north of Atlit (Atlit II, Atlit III), where the beaches are wide, with sand dunes at the backshore. In some of the sites (Ashkelon, Jisr az-Zarka, Habonim-Dor, Atlit I, Akko, and Shavei Tzion Beit HaEmek stream outlet), the contact points between the BR outcrop base and the rocky substrate were detected, showing that the in-situ BR outcrop is positioned directly or partly on hard substrate. In the Atlit I and Atlit II sites, the substrate consists of dark clay, as was shown in previous studies, e.g., [71] and validated by excavation. At the other sites, no contact points with hard bedrock were detected, and the BR outcrops in those sites lie on unconsolidated sediment substrate or partly on unconsolidated substrate and partly on hard substrate (most probably, drilling or excavation would be necessary for final determination of this). In the Gador 1 site the crest of the BR intertidal band was punched with a drill, showing it is about 0.8 m thick, and validating its substrate identification as unconsolidated sediment. Some of the sites are on beaches that are affected significantly by construction (Netanya I, Netanya II, Mikhmoret I, Binyamin Cove and Angler Cove, Akko, and Shavei Tzion), and others are in wild beaches, where the effect of modern construction is very low or not significant (Arsuf, Gador). Overall, dimensions of the observed BR outcrops are consistent with those presented in former studies, e.g., [40], although our measurements include outcrops which are wider (55 m in Habonim-Dor I, 60 m in Shavei Tzion Beit HaEmek Stream) than those previously reported. Submerged detached BR slabs were observed and sampled at Ashkelon II (about 3 m water depth), Mikhmoret I (about 2 m water depth), and Gador I (about 2 m water depth). These BR bands are parallel to the in-situ intertidal BR bands and are found up to 30 m from it. Unconsolidated sediment surface is present between the submerged and intertidal BR bands.

The main clastic components in all sites were identical to the unconsolidated sediment of the beach in each site. Shells and shell fragments were detected in all outcrops. Quartz sand is a major component in sites near Ashkelon, which are located closest to its main source. The relative amount of kurkar fragments (mostly granule size) is larger along the Sharon coast, where a continuous coastal cliff is present backshore along most of the shoreline. Biogenic sand is the main component of BR outcrops of Akko and Shavei Tzion that are located north of Haifa Bay. Inclusions of archaeological material (e.g., ceramic and marble pieces) are found in consolidated BR, e.g., at Atlit II and Akko (Table 1: 27 and 29).

## 4. Discussion

### 4.1. Classification of Beachrock Morphological Structures

The morphological features in the BR outcrops studied are presented here as a generalized conceptual model, based on all sites measured (Figure 10). These main characteristics are distinctive and common in BR outcrop morphology, regardless of the differences in beach type, coastline geometry, sedimentary composition, and the intensity of anthropogenic effect on the coastline. These morphologies described below, are related to the BR formation process, its later exposure due to erosion, and consequent weathering.

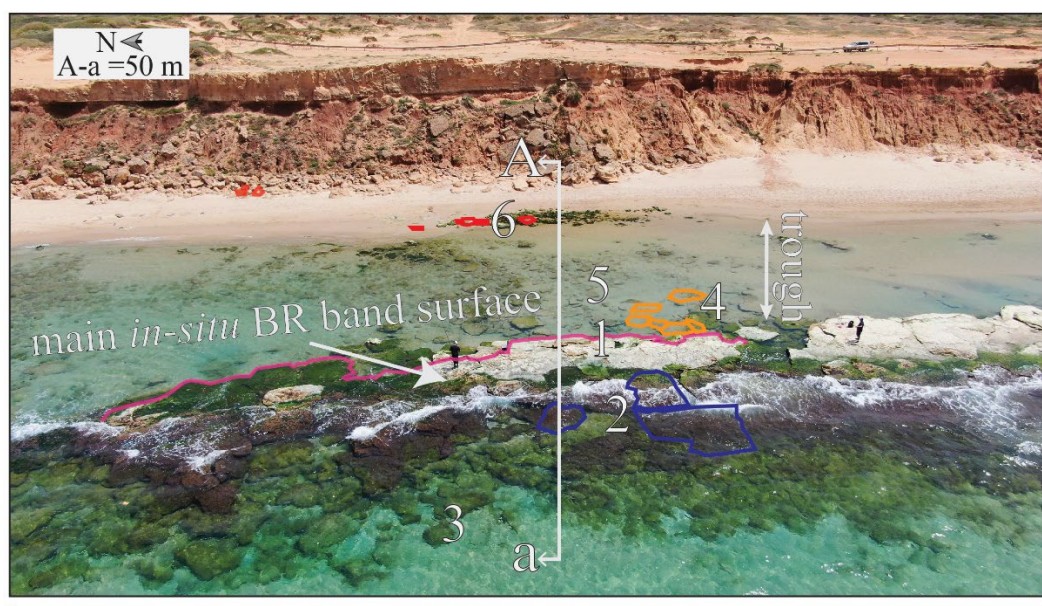

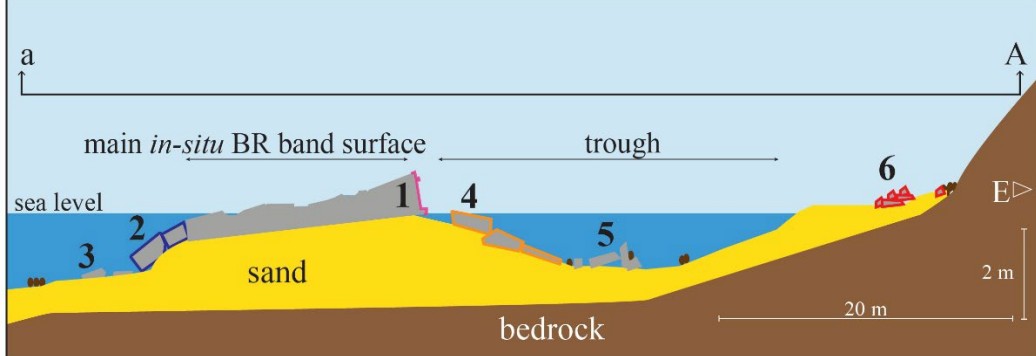

**Figure 10.** Aerial view (upper) and schematic lateral cross-section (lower) of BR outcrop: showing typical erosion morphological features occurring on intertidal BR outcrops. Numbers represent the morphological features (**1**: main band of in-situ BR outcrop, **2**: tilted detached BR beds which underwent settling facing the sea side, **3**: detached BR plates which settled near the outcrop sea side front; **4**: detached BR plates which dropped and settled near the outcrop's land side front, **5**: detached BR plates concentrated inside the trough, **6**: detached BR plates concentrated at backshore).

In this paper we refer first to features appearing on the in-situ BR band, and then the features characterizing its sea side and land side extremities: (1) development of a micro-cliff at the front of the in-situ BR facing the backshore (Figure 10: 1), and creation of a trough at the land side front of the micro-cliff (due to flow of backwash waters along BR outcrop backshore face); (2) and settling of the marginal beds of BR facing seawards due to erosion and scouring. The detached BR blocks are scattered on the sea bottom in a steeper declination towards the sea than the surface of the main in-situ BR (Figure 10: 2); (3) BR plates are scattered around the sea side margin of the main in-situ BR (Figure 10: 3); (4) BR beds drop off at the land side of the outcrops (Figure 10: 4); (5) BR plates are scattered around the land side margin of the main in-situ BR (Figure 10: 5); (6) BR plates heap

up backshore (Figure 10: 6); abrasion and solution marks appear on the outcrop surface (including furrows, bumps, holes, slots, lithological notches, cracks, and pits).

The current study classifies distinctive erosional structures of BR outcrops associated with the lithological configuration of the bedrock and the coastal sediment lens in each different scenario.

Scenario (A): The BR lens is embedded on the hard substrate (Figure 11A). The erosional processes and morphological features include: (1) Erosion of the sea side is in part due to wave pounding. Since the base layer of the BR package is situated directly on a hard subsurface, no severe erosion or scouring occurs, projecting ledges are left unsupported and collapse, remaining on the exposed subsurface. Abrasion occurs on the exposed surface of the BR. Consequently, abrasion features of the surface deepen and become failure points, and parts of the BR outcrop are separated from the main band. (2) Erosion at the land-facing front includes forming notches at the BR outcrop front, resulting in collapsing beds. Large plates remain near their former location, and a micro-cliff develops as the front retreats; (3) and plates break and are thrown towards the backshore.

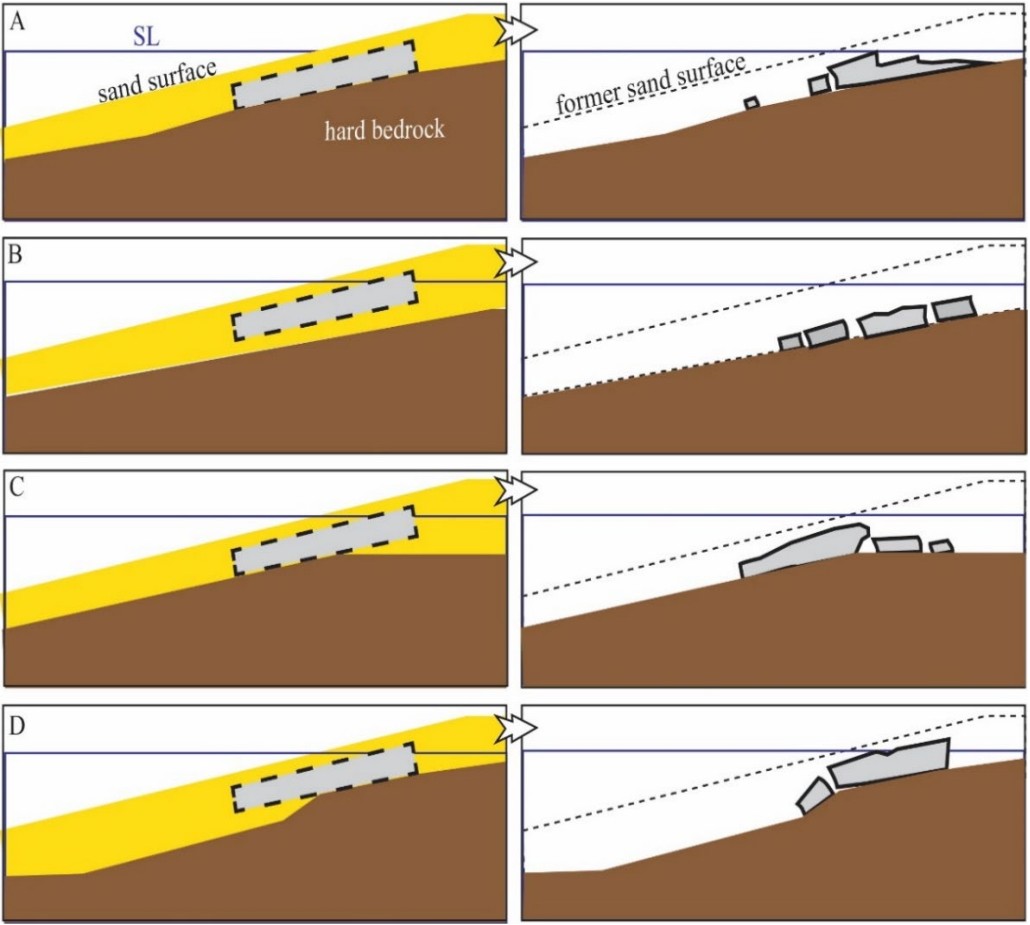

**Figure 11.** Erosion patterns of BR outcrops arranged by different underlying lithologies, assuming one stage of formation, and stable sea level. The illustrations on the left show initial stage: formation under sediment coverage. The illustrations on the right show the situation after erosion of the loose sediment and exposure of the BR outcrop; (**A**) (Scenario A) BR lens on hard substrate; (**B**) (Scenario B) BR lens on loose substrate; (**C**) (Scenario C) BR lens on hard surface on its sea side part and loose sediment on its land side part; (**D**) (Scenario C) BR lens on hard sub-surface on its land side part and loose sediment on its sea side part.

Scenario (B): The BR lens is embedded on loose substrate (Figure 11B). The erosion processes and morphological features include: (1) Erosion and scouring at sea side part. Eventually, after the disappearance of most of the unconsolidated sediments under the BR,

its parts undergo settling and make contact with the hard bedrock below; (2) erosion of the land side part as the water line moves landward, including plates dropping and creation of a micro-cliff at the front; (3) and plates scattered near outcrop margins.

Scenario (C): The BR lens lies partly on hard substrate and partly on loose substrate. The erosion processes and morphological features include: (1) BR lens situated on loose base on its land side and hard sub-layer on its sea side (Figure 11C) includes, after the disappearance of covering sediments, large plates dropping along the land-facing front and detached plates situated close to the outcrop ledge. BR plates are found on the ground in positions that follow the beach profile angle at their location; (2) and BR lens on hard sub-base surface on the land side part and loose sediment on the sea side part (Figure 11D) including erosion on the sea side part, resulting in gradual breaking of the front and creation of a collapse apron consisting of plates dipping towards the sea. The part of the lens which is situated on a hard surface undergoes abrasion and is characterized by major development of surface erosional features, such as troughs, a micro-cliff, projecting ledges, and cracks. Many unattached blocks are scattered on the surface. The BR layer becomes narrower as it approaches the bedrock; in all cases of alternative properties of the sub-layer, plates break and are thrown onto the backshore (such as A (3) B (3)).

### 4.2. Beachrock Exposures: A Proxy for Sandy Beach Erosion at the Mediterranean Coast of Israel

Beachrock occurrence is usually associated with the intertidal zone of microtidal coasts, in warm environments, with sufficient meteoric water (but also in arid climates). The formation of BR is a consequence of steady, calm sea conditions that enable the build-up of a significant sediment lens during a dynamically stable environment of at least a few years. Several studies of BR exposures along the Israeli coast have been published [40,62,64–68,73]. Optically stimulated luminescence (OSL) ages obtained from some of the outcrops [67,68] point to a time frame of their initial formation between 2000 and 800 years BP.

Once exposed, as described in previous studies, e.g., [40,70,74], the BR outcrop is subjected to weathering and abrasion, resulting in various morphological features. Beachrock outcrops are also subjected to chemical and biological weathering, but the influence of erosion caused by water force and gravitation on the outcrop's morphology seems to be more significant. Water swashes and wave pounding on the BR surface remove sediments above and around the outcrop's ledges, scouring and moving loose sediment, undermining the in-situ deposit, and deepening cracks and channels.

Erosion, in relation to coastal geomorphology, is a term used to describe shoreline retreat. Naturally, rocky headlands are predominantly subjected to high energies and erosion, while deposition occurs in low energy areas within bays [75,76]. Coastal erosion is a global phenomenon that can damage man-made physical structures, and it has economic influence on human activities, especially coastal communities. Therefore, the study of erosion is crucial for planning and managing coastal resources [77,78].

Beachrock can be used as an important indicator for coastal erosion at various time scales: seasonal changes can be detected by direct observation, decadal changes can be detected using aerial photography, and historical or prehistoric changes can be investigated based on geomorphological, sedimentological, geochronological, and archaeological tools, (e.g., [8–13,28]). The current study describes the morphology of selected BR deposits formed during the late Holocene, at scales of multi-site study, and investigates their erosion process and connection to beach evolution. The occurrence of in-situ BR beds exposed along the studied parts of the coast of Israel allows the identification of areas where the coastline has retreated, and erosional evidence of BR is observed. The observed morphology in the study area suggests that a wider sediment lens was present in the historical times along the Sharon coast, but has been eroded. BR outcrops at the rocky beaches of Habonim and Dor, Nun Cove, Akko, and Shavei Tzion indicate that sandy beaches developed in these places in the past, for a few decades at least. BR outcrops up to a few km long exposed along sandy beaches (e.g., Ashkelon, Hadera, and Mikhmoret), indicate that these beaches were wider during the initial formation stage of the BR.

Coastal erosion results from natural processes, such as the action of wind, waves, currents, and sea-level changes [76] and anthropogenic action (e.g., sand quarrying and marine constructions). Considering that the climate and wave regime along the Mediterranean coast of Israel have not changed significantly since the formation of BR [54,78]; nor have the composition and amount of sediments deposited in the area [53], and given that no significant tectonic event has occurred there in late Quaternary [44], there are few relevant causes of sediment erosion in relation to BR exposure. These are: (a) local changes in the shoreline due to nearby coastal and marine constructions (factor over at least 50 years); (b) sand quarrying on beaches (over the last 100 years, until the enactment of the "Zifzif Law" in the 1963s) [40]; (c) extreme wave storms that can transport large amounts of sediment in a short time (factor in the last thousands of years); and (d) sea level fluctuations $\pm$ 1 m during the last 4000 years [49,79].

The occurrence of large BR outcrops, both on undisturbed beaches and on beaches subjected to significant anthropogenic activity, indicate that modern construction is not the only factor that causes erosion. Sand quarrying may explain BR exposure in some cases, e.g., in Shavei Tzion [40]. However, further comprehensive studies along the entire Israeli coast are required to estimate its extent. In some cases, significant parts of the BR outcrops were quarried and relocated as an architectural element (e.g., Ashdod, Palmachim, Nun Cove, and Shavei Tzion) [40,73]. The occurrence of large, detached BR sections at 2–3 m water depth (Ashkelon: Figure 3D; Mikhmoret, Gador: Figure 6C; Habonim—Dor: Figure 7D, Shavei Tzion) does not necessarily indicate a lower sea level during the BR formation or post-deposition tectonic subsidence. Sediment erosion, scouring and settling processes can explain their current location. Detached BR plates observed in the study area at up to 3 m above sea level have been thrown by storm waves towards the backshore, and do not necessarily indicate higher sea level at the time of formation.

### 5. Conclusions

1. In-situ, beachrock outcrops are subjected to marine weathering and erosion, characterized by morphological features typical of different underlying lithologies.
2. Beachrock outcrops along the Israeli Mediterranean coast, located at various beach types and sites differing in coastline geometry, sediment composition, and the intensity of anthropogenic impact on the coastline, are characterized by similar typical distinctive morphological features.
3. The occurrence of beachrock outcrops along considerable parts of the Mediterranean coast of Israel points to an erosion phase of sediment that once covered the beaches when the beachrock was formed.
4. The occurrence of large BR sections on the seabed (up to 4.5 m bsl) and above current sea level (up to 3 m asl) does not necessarily indicate different sea levels at the time of their formation, nor tectonic changes. Erosion, scouring, and settling processes can explain their location on the sea bottom, while wave activity is responsible for their location in the foreshore and backshore.

**Author Contributions:** Conceptualization, A.B., E.G., D.Z. and R.B.; investigation and resources, A.B., D.Z., R.B. and E.G.; writing—original draft preparation, A.B.; writing—review and editing, E.G., D.Z., A.B. and R.B.; visualization, A.B., E.G. and D.Z.; supervision, D.Z. and R.B. All authors have read and agreed to the published version of the manuscript.

**Funding:** This research received no external funding.

**Institutional Review Board Statement:** Not applicable.

**Informed Consent Statement:** Not applicable.

**Acknowledgments:** The authors are grateful to Reuven Bar for his valuable assistance with field work. The authors thank John Tresman for the English editing of this manuscript.

**Conflicts of Interest:** The authors declare no conflict of interest.

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
