# Peer review of "Beachrock Morphology along the Mediterranean Coast of Israel: Typological Classification of Erosion Features"

_jmse, doi:10.3390/jmse10111571_

Round 1

Reviewer 1 Report

Dear Author,

The theme presented “Beachrock Morphology along the Mediterranean Coast of Israel: Typological Classification of Erosion” is relevant to the understanding of relative sea level - RSL on Mediterranean coast during the Holocene. However, the manuscript is very descriptive and little analytical in terms of genesis of these rocks. It focused on recording the location of outcrops and their erosion patterns.

The methods is inconsistent for not using analytical techniques, among these, petrographic analysis, stable isotopes, cathodoluminescence and radiocarbon dated, among others. From a genetic perspective, beachrocks can be divided into four categories according to the process of lithification: (a) precipitation of highly magnesian calcite and aragonite from seawater due to high temperature, oversaturation and/or CaCO3 evaporation; (b) precipitation of calcite and aragonite from mixture of groundwater and seawater; (c) precipitation of magnesian calcite and aragonite by CO2 degassing from sediment and pore water; (d) precipitation of micrite calcium carbonate as by product of microbiologic activity (Mauz et al., 2015 and Malta et al 2021). These genetic processes may support the typological classification presented by the authors according to the references below:

·       Mauz B., Vacchi M., Green A., Hoffmann G., Cooper A., 2015. Beachrock: A tool for reconstructing relative sea level in the far-field. Mar Geol 362: 1- 16.

·       Malta, J.V; Castro, J.W.A; Cabral, C.L; Fernandes, D; Cawthra, H. C. 2021 Genesis and age of beachrocks on the Rio de Janeiro coastline, Southeast - Brazil. Mar Geol.                                                         v. 442, p. 106649-106661.

I recommend a thorough review so that the manuscript can be published in the Journal of Marine Science and Engineering according to the annotations below:

·      Broadly, in the last paragraph of Introduction (1), inform bibliographic references on typological and classification of the beachrocks in other parts of world. I also emphasize the importance of formulating a hypothesis about the occurrence of beachrock along the Mediterranean coast of Israel.

·      In the method (2), describe in detail all the techniques used in relation to the execution of beach profiles, topographical maps (scale), aerial images (flight dates). Try to fit the information obtained in the genetic classification presented by Mauz et al (2015) and Malta et al (2021).

·      In the results obtained, the morphological description of the outcrops is very extensive and confusing. We recommend that most of information be placed in table 1.

·      Discussion of results obtained should be further explored.

·      The quality of figures are very good. I recommend that they are grouped according with the description of first paragraph of  results (3)

·      The conclusion presented is not satisfactory, explore the issues raised in the abstract a little more.

Therefore, we recommend major modifications so that the manuscript can be published.

 Sincerelly,

Author Response

We are grateful for the comments of reviewer #1 and a detailed reference document is attached

Reviewer 2 Report

Dear Authors,

I have enjoyed reviewing your paper ‘Beachrock Morphology along the Mediterranean Coast of Israel: Typological Classification of Erosion’ and have attached an annotated PDF with specific comments, as well as general comments below.

In the methods, drilling is mentioned at the end, but what type of drilling and how deep did your sample penetrate? What did you use the drilled core material for specifically?

General comments:

Throughout the text: when referring to the unconsolidated sediment (recent beach deposits) you call it sand. But is the grain size fraction only sand with no gravel or silt, for example? I suggest a find-replace on ‘sand’ and replace this with ‘unconsolidated sediment’ or ‘modern sediment’ or ‘sediment’.

Instead of ‘strap’ I suggest using either band or strip, or outcrop, or unit.

Instead of ‘sub-basing’ please use substrate, or underlying.

Exposure, not exposer throughout

Please do not use the word ‘massive’ if you are referring to very large. Massive is fine if you mean homogenous, but if referring to size then this needs to be quantified with measurements, even if they are approximate.

For the results, your archaeology incorporated in the beachrock, and available as a tool for dating, is wonderful. I feel that a table summing up these results and including these dates, where you can constrain time periods, will be helpful. Consider an overview table, perhaps? (like the one you have as an appendix).

I also felt that the table in the appendix / supplementary material is not too big to put in the main text. I suggest inclusion in the results.

At the start of your discussion, I think it will be useful to the reader to frame these ages too. For example, saying that they were (at least, some) deposited between xx and xx years before present.

In the section around line 533, this can be tabulated too. You present various scenarios for erosion of beachrock slabs and it is a bit confusing to absorb it how it is currently written with the A, B, C and 1, 2, 3, etc. Please either clarify the text, or consider a table?

In the paragraph of line 582, there is a comment that implies that the presence of beachrock can inform on the size of the beach on which it formed, but I would not agree with this. The presence of beachrock doesn't provide a proxy for how much sediment was there when it formed on the contemporary beach. I suggest leaving this out. For sure, beachrock indicates that a sandy beach was present, but not the characteristics of that beach.

At the end of the discussion and in the conclusions, there is a statement about coastal stability being inferred from beachrock. However, I did not see thorough results and discussion around the ages of these beachrock units. Some were constrained in time with archaeological evidence, but does this apply to all? i.e., can you do this for every unit that you mapped? If so, this needs more attention in the manuscript and perhaps a table of Region – Beachrock Unit – Inferred age (numbers) – Justification (here, say how it allowed you to date it with the relative associations).

If you do not have enough data to talk to coastal stability, then I think this whole part of the discussion can be left out. If your contributions on morphology and erosion add to what is currently known, then that is probably enough for this paper.

In general, I found the discussion to be a little unclear. What I expected, after reading the results, was a discussion kind of like:

1.     Comparison across the regions described

2.     Beachrock morphology, as determined by this study

3.     Erosional features of beachrock

4.     Comparison to the literature on this (what is new, what is different, etc).

I think after a re-read, and implementing revisions, you may be well positioned to reorganise the discussion into a very clear logic flow of your findings and what those mean.

I think this is a valuable contribution to the beachrock literature and hope to see your work published!

Kind regards.

Author Response

We are grateful for the comments of reviewer #2 and a detailed reference document is attached

Reviewer 3 Report

The study presents an overview of the dynamic transformations of the coast of Israel where BR outcrops, and tries to establish erosion patterns for the morphologies found.

Line 54 Figure 1: In the presented figure, the drawing shows that the BR are formed without contact with the antecedent substrate (bedrock). Is there any evidence of this?

Line 89: Is there any tidal influence off the coast of Israel? Write a sentence describing whether or not tidal influence occurs and their respective values.

Line 104: Was the thickness measured through perforations, or is this what comes out on the face of the beaches?

Line 127: Are aerial images large format aerial photographs (aerophotogrammetric surveys?) If taken with UAVs (drone...etc)? Describe the type of UAV and which are oblique. Aerial images, even oblique, are important in geomorphological characterizations depending on the scale.

Line 128 to Line 132: Elevation error of up to 30 cm may no longer characterize erosion or deposition. Another point in relation to the method used in the topography would be the level reference (average relative level in the sea)? What was adopted? Are there level references in the places that carried out the topographic surveys?...Were they corrected with the tide?

Line 137-138: Was the petrological characterization of the samples macroscopic? Performed in the laboratory with a binocular magnifying glass? How many samples??

Line 166: What kind of archaeological remains?

Line 184: Waterline? I believe that the term coastline” or “shoreline” would be more correct

Line 188 (figure 3): In the graphic representation of the profile, what was the vertical exaggeration? Describe in legend. The photographs lack scales (3C, 3D, 3E). Possible solution, mention in the caption.

Line 224: Figure 4 needs to calculate the vertical exaggeration for profiles B and D. Comparing Figure B and D there should be an error in the representation of measurements of length and height of figure D (proportion in relation to profile B). Also relative to the thickness of bedrock and sand in the drawing) In the drawing of figure D does it show a bedrock clast welded to the BR? Is that it, or is it a loose clast?

Line 238 Waterline? I believe that the term “coastline” or “shoreline” would be more correct

Line 253: Figure 5 (Set the vertical exaggeration between 5B, 5D, 5F) Scale in Figure 5C. Letter d in Figure 5D is not represented in Figure 5C.

Line 253: Figure 6(Set the vertical exaggeration between 6A) Scale in figure 6B and 6C.

Line 351: Why wasn't there a representative drawing and photos for this section? (Hadera Segment)

Line 405: Figure 7(Set the vertical exaggeration between 7B) Scale in photographs 7C, 7D, 7E. It is not possible to read clearly what is written about photographs 7C, 7D, 7E.

Line 412: Figure 8 (Set the vertical exaggeration between 8B).

Line 432: Here appears the first occurrence of BR that is consistent in descriptive terms., For if archaeological fragments occur, this certainly distinguishes them from outcropping rocks (bedrocks). Atlit North

Line 447: Figure 8 (Set the vertical exaggeration between 8C).

The results of the petrological descriptions of the beach rocks could be illustrated with photos along with table descriptions, so they could be classified in a more organized way. I mention this because there are descriptions about the BRs, but there is no illustration, pointing/showing similarities and/or differences between them and if there is a similarity and/or difference regarding the sediments that make up the beaches as well as the different types of bedrocks. Before pointing out the erosive processes, the formation of these BRs would be minimally discussed, with what is described in the literature (citations). If there are differences in bedrock along the coast, there are also differences between the sediments that make up the beaches. Were these BR formed during the Holocene? What are the main differences/similarities between BRs?

A suggestion would be to propose a single figure for the discussion that would help to understand the differences / similarities along the coast, just showing the profiles, simplified and at the same scale, superimposed or one below the other. This figure would demonstrate the differences that occur between them and corroborates with figure 10 which would be the possibilities.

Line 591: Coastal erosion is being addressed as a factor at what time scale? Throughout the Holocene? On a historical and/or decadal scale, or on a seasonal scale? For each scale there should be indicators (ages in the case of the Holocene) aerial photographs if it's a decadal...showing the retreat of the coastline? Before and after if it's seasonal...

Line 598-604: A point that can be considered? Is there a natural contribution of sediments through rivers and/or the continental shelf? Could the lack of natural sedimentary input also be a factor that contributes to coastal erosion?

Line 605-616- Different radiocarbon ages and/or LOE already published in BR that surface on beaches and/or in submerged BR doesn't help to explain the processes of coastal retreat and the scale that is occurring?

Finally, I believe that the article has the potential to be published, however some adjustments can make it more interesting. Erosion-related factors and causes must be addressed taking into account different time scales. Coastal retreat may have overlapping factors, and BR ages can help a lot in this process.

Author Response

We are grateful for the comments of reviewer #3 and a detailed reference document is attached

Round 2

Reviewer 1 Report

Dear Author,

The corrections presented are satisfactory. Virtually all of our requirements were well executed. 

Reviewer 2 Report

Dear Authors, 

Thank you for addressing the recommended revisions - I am happy to accept your manuscript in this updated form. 

Look forward to seeing it published! 

Reviewer 3 Report

Dear!

Thanks for answering the reviews!

The article is better.

Att

Reviewer